# PARAMETERIZED PROJECTED BELLMAN OPERATOR

## ABSTRACT

The Bellman operator is a cornerstone of reinforcement learning, widely used in a plethora of works, from value-based methods to modern actor-critic approaches. In problems with unknown models, the Bellman operator requires transition samples that strongly determine its behavior, as uninformative samples can result in negligible updates or long detours before reaching the fixed point. In this work, we introduce the novel idea of obtaining an approximation of the Bellman operator, which we call projected Bellman operator (PBO). Our PBO is an operator on the parameter space of a given value function. Given the parameters of a value function, PBO outputs the parameters of a new value function and converges to a fixed point in the limit, as a standard Bellman operator. Notably, our PBO can obtain approximated repeated applications of the true Bellman operator, as opposed to the sequential nature of the standard Bellman operator. We show how to obtain PBOs for representative classes of RL problems, and how to approximate it resorting to neural network regression. Eventually, we propose an approximate value-iteration algorithm to learn PBOs and empirically evince how it can overcome the limitations of classical methods, opening up multiple research directions as a novel paradigm in reinforcement learning.

## 1 INTRODUCTION

Value-based reinforcement learning (RL) is a popular class of algorithms for solving sequential decision-making problems with unknown dynamics (Sutton & Barto, 2018). For a given problem, value-based algorithms aim at obtaining the most accurate estimate of the expected return from each state, i.e., a value function. For instance, the well-known value-iteration algorithm computes value functions by iterated applications of the Bellman operator (Bellman, 1966), of which the true value function is the fixed point. Although the Bellman operator can be applied exactly in dynamic programming, it needs to be estimated from samples *at each application* when dealing with unknown models of RL problems, i.e., empirical Bellman operator (Watkins, 1989; Bertsekas, 2019). Intuitively, the dependence of value iteration on the samples has an impact on the efficiency of the algorithms and on the quality of the obtained estimated value function, which becomes accentuated when solving continuous problems that require value-based methods with function approximation, e.g., approximate value iteration (AVI) (Munos, 2005; Munos & Szepesvári, 2008). Moreover, in AVI approaches, costly function approximation steps are needed to project the output of the Bellman operator back to the considered action-value functional space.

In this paper, we tackle these limitations by introducing the novel approach of using samples to obtain a new operator, which we call *projected Bellman operator* (PBO). Our PBO is a function $\Lambda : \Omega \to \Omega$ defined on parameters $\omega \in \Omega$ of the value function. Contrarily to the standard (empirical) Bellman operator, which uses action-value functions $Q_{\omega_k}$ to compute targets that are then projected to obtain $Q_{\omega_{k+1}}$, our PBO uses the parameters of the action-value function to compute updated parameters $\omega_{k+1} = \Lambda(\omega_k)$ directly (Figure 1). The crucial advantages of our approach are twofold: (i) the output of PBO always belongs to the considered action-value functional space, avoiding, therefore, the costly projection step typical when using the Bellman operator, and (ii) once learned, PBO is applicable for an arbitrary number of iterations without using further samples, as visualized in Figure 2. Starting from initial parameters $\omega_0$, AVI approaches obtain consecutive approximations of the value function $Q_{\omega_k}$ by

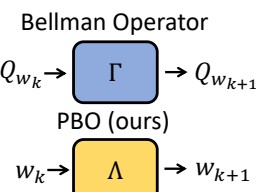

Bellman Operator

$Q_{w_k} \to$ Γ $\to Q_{w_{k+1}}$

PBO (ours)

$w_k \to$ Λ $\to w_{k+1}$

Figure 1: PBO operates on value function parameters.

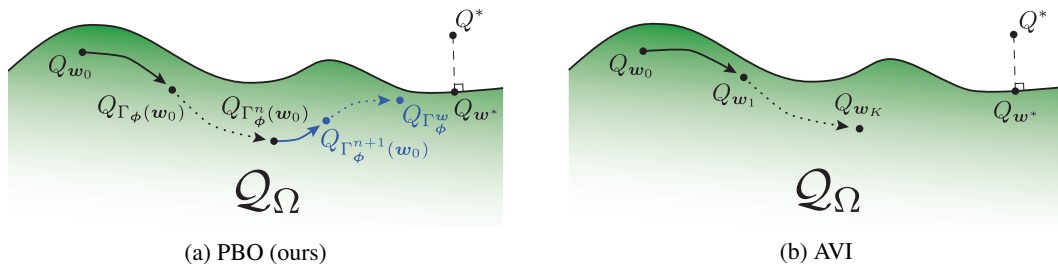

Figure 2: Behavior of our PBO and approximate value iteration (AVI) in the space of value functions. $Q^*$ and $Q_{\omega^*}$ are respectively the optimal value function and its projection on the parametric space.

applying the Bellman operator iteratively over samples (Figure 2b). Instead, our PBO makes use of the samples to learn the operator only. Then, starting from initial parameters $\omega_0$, PBO can produce a chain of updated parameters of arbitrary length (blue lines in Figure 2a) without requiring further samples. In the following, after formally introducing PBO and a novel algorithm for value estimation based on it, we analyze its advantageous properties for different classes of problems. Thus, our contribution is threefold: (i) we introduce the notion of projected Bellman operator (PBO); (ii) we show how to derive different PBOs according to the class of problems at hand; (iii) we develop a novel algorithm for value estimation based on PBO and show its advantages over related baselines on several RL problems.

## 2 RELATED WORKS

Our work is, to the best of our knowledge, the first attempt to obtain a variant of the Bellman operator that acts on the parameters of action-value functions. Nevertheless, several works in the literature proposed variants of the standard Bellman operator to induce some desired behavior.

**Variants of the Bellman operator** are widely studied for entropy-regularized MDPs (Neu et al., 2017; Geist et al., 2019; Belousov & Peters, 2019). The softmax (Haarnoja et al., 2017; Song et al., 2019), mellowmax (Asadi & Littman, 2017), and optimistic (Tosatto et al., 2019) operators are all examples of variants of the Bellman operator to obtain maximum-entropy exploratory policies. Besides favoring exploration, other approaches address the limitations of the standard Bellman operator. For instance, the consistent Bellman operator (Bellemare et al., 2016) is a modified operator that addresses the problem of inconsistency of the optimal action-value functions for suboptimal actions. The distributional Bellman operator (Bellemare et al., 2017) enables to operate on the whole return distribution, instead of its expectation, i.e., the value function (Bellemare et al., 2023). Furthermore, the logistic Bellman operator uses a logistic loss to solve a convex linear programming problem to find optimal value functions (Bas-Serrano et al., 2021). Finally, the Bayesian Bellman operator is a method in Bayesian RL to infer a posterior over Bellman operators centered on the true Bellman operator (Fellows et al., 2021). We point out that our PBO can be seamlessly applied on an arbitrary variant of the standard Bellman operator with just minor adaptations.

**Operator learning.** Literature in operator learning is mostly focused on supervised learning, with methods for learning operators over vector spaces (Micchelli & Pontil, 2005) and parametric approaches for learning non-linear operators (Chen & Chen, 1995), with a resurgence of recent contributions in deep learning. For example, Kovachki et al. (2021) learn mappings between infinite function spaces with deep neural networks, or Kissas et al. (2022) apply an attention mechanism to learn correlations in the target function for efficient operator learning. The literature about operator learning is much larger, but we consider it out of the scope of this work that, to the best of our knowledge, is the first to deal with the original problem of operator learning in RL.

## 3 PRELIMINARIES

We consider discounted Markov decision processes (MDPs) $\mathcal{M} = \langle \mathcal{S}, \mathcal{A}, \mathcal{P}, \mathcal{R}, \gamma \rangle$, where $\mathcal{S}$ is a measurable state space, $\mathcal{A}$ is a measurable action space, $\mathcal{P} : \mathcal{S} \times \mathcal{A} \rightarrow \Delta(\mathcal{S})$[1] is the transition

---

[1] $\Delta(\mathcal{X})$ denotes the set of probability measures over the set $\mathcal{X}$.

kernel of the dynamics of the system, $\mathcal{R} : \mathcal{S} \times \mathcal{A} \to \mathbb{R}$ is a reward function and $\gamma \in [0, 1)$ is a discount factor (Puterman, 1990). A deterministic policy $\pi : \mathcal{S} \to \mathcal{A}$ is a function mapping a state to an action, inducing a value function $V^\pi(s) \triangleq \mathbb{E}\left[\sum_{t=0}^\infty \gamma^t \mathcal{R}(S_t, \pi(S_t))|S_0 = s\right]$, i.e., expected cumulative discounted reward following policy $\pi$. Similarly, the action-value function $Q^\pi(s, a) \triangleq \mathbb{E}\left[\sum_{t=0}^\infty \gamma^t \mathcal{R}(S_t, A_t)|S_0 = s, A_0 = a, A_t = \pi(S_t)\right]$ is the expected discounted cumulative reward executing action $a$ in state $s$, following policy $\pi$ thereafter. RL aims to find an optimal policy $\pi^*$ yielding the optimal value function $V^*(s) \triangleq \max_{\pi:\mathcal{S}\to\mathcal{A}} V^\pi(s), \forall s \in \mathcal{S}$ (Puterman, 1990).

The (optimal) Bellman operator $\Gamma^*$ is a fundamental tool in RL for obtaining optimal policies, and it is defined as

$$(\Gamma^* Q)(s, a) \triangleq \mathcal{R}(s, a) + \gamma \int_\mathcal{S} \mathcal{P}(\mathrm{d}s'|s, a)V^*(s'), \quad \forall (s, a) \in \mathcal{S} \times \mathcal{A}. \tag{1}$$

It is well-known that Bellman operators are contraction mappings in $L_\infty$-norm, such that iterating over an action-value function leads to the fixed point $\Gamma^* Q^* = Q^*$ in the limit (Bertsekas, 2015). We consider the use of function approximation to represent value functions and denote $\Omega$ the space of their parameters. Additionally, we define $\mathcal{Q}_\Omega = \{Q_{\boldsymbol{\omega}} : \mathcal{S} \times \mathcal{A} \to \mathbb{R}|\boldsymbol{\omega} \in \Omega\}$ as the set of value functions representable by parameters of $\Omega$.

## 4 PROJECTED BELLMAN OPERATOR

The application of the Bellman operator in RL requires transition samples (Equation 1) (Munos, 2003; 2005; Munos & Szepesvári, 2008). The impact of this requirement is twofold: (i) the behavior of the Bellman operator strictly depends on the quality of the samples, thus a poor sample selection can result in negligible updates and long detours before convergence; (ii) the direction of the update is determined exclusively by the current samples, meaning that the empirical Bellman operator has no memory of previously observed samples (Bellemare et al., 2016; Agarwal et al., 2019; Fellows et al., 2021). Ideally, we want to obtain an operator that mirrors the behavior of the true Bellman operator while being independent of the transition samples and extrapolating from previous experience. Thus, we introduce the idea of an alternative operator, which we call *projected Bellman operator* (PBO), defined as follows.

**Definition 1.** *Let $\mathcal{Q}_\Omega = \{Q_{\boldsymbol{\omega}} : \mathcal{S} \times \mathcal{A} \to \mathbb{R}|\boldsymbol{\omega} \in \Omega\}$ be a function approximation space for the action-value function, induced by the parameter space $\Omega$. A projected Bellman operator (PBO) is a function $\Lambda^* : \Omega \to \Omega$, such that*

$$\Lambda^* \in \underset{\Lambda:\Omega\to\Omega}{\arg\min} \, \mathbb{E}_{(s,a)\sim\rho, \boldsymbol{\omega}\sim\nu} \left[ (\Gamma^* Q_{\boldsymbol{\omega}}(s, a) - Q_{\Lambda(\boldsymbol{\omega})}(s, a))^2 \right]. \tag{2}$$

*for some state-action distribution $\rho$ and parameter distribution $\nu$.*

Note that the symbol $\Gamma^*$ is the regular optimal Bellman operator on $\mathcal{Q}_\Omega$. Conversely, PBO is an operator $\Lambda^*$ acting on the space of parameters $\Omega$. All proofs of following results can be found in Appendix A.

### 4.1 FUNCTION APPROXIMATION FOR PROJECTED BELLMAN OPERATORS

In the following, we show that PBO can be expressed in closed-form for representative classes of RL problems: (i) finite MDPs, (ii) low-rank MDPs (Agarwal et al., 2020; Sekhari et al., 2021), and (iii) linear quadratic regulation problems (Bradtke, 1992; Pang & Jiang, 2021).

#### 4.1.1 FINITE MARKOV DECISION PROCESSES

Let us consider finite state and action spaces of cardinality $N$ and $M$ respectively, and a tabular setting with $\Omega = \mathbb{R}^{N \cdot M}$. Intuitively, there is a trivial bijection between $\mathcal{Q}_\Omega$ and $\Omega$, which allows us to write the parameters of the action-value function as $Q$ instead of $\boldsymbol{\omega}$.

**Proposition 1.** *The PBO exists and it is equal to the optimal Bellman operator*

$$\Lambda^*(Q) = R + \gamma P \max_{a \in \mathcal{A}} Q(\cdot, a). \tag{3}$$

Note that, since the optimal Bellman operator is a $\gamma$-contraction mapping for the sup-norm, PBO for finite MDPs (Equation 3) is also a $\gamma$-contraction mapping.

### 4.1.2 LOW-RANK MARKOV DECISION PROCESSES

Low-rank MDPs is a class of problems with two feature maps $\boldsymbol{\sigma} : \mathcal{S} \times \mathcal{A} \to \mathbb{R}^d$ and $\boldsymbol{\mu} : \mathcal{S} \to \mathbb{R}^d$, such that $\mathcal{P}(s'|s, a) = \langle \boldsymbol{\sigma}(s, a), \boldsymbol{\mu}(s') \rangle$ and $\mathcal{R}(s, a) = \langle \boldsymbol{\sigma}(s, a), \boldsymbol{\theta} \rangle$, for all $(s, a, s') \in \mathcal{S} \times \mathcal{A} \times \mathcal{S}$ and for $\boldsymbol{\theta} \in \mathbb{R}^d$. We assume, without loss of generality, that for all $(s, a) \in \mathcal{S} \times \mathcal{A}, \|\boldsymbol{\sigma}(s, a)\|_1 \leq 1$ and $\max\{\|\boldsymbol{\mu}(s)\|_1, \|\boldsymbol{\theta}\|_1\} \leq \sqrt{d}$. We set the space of action-value functions to be the space of linear functions in the parameters, i.e., $\mathcal{Q}_\Omega = \{\langle \boldsymbol{\sigma}(\cdot, \cdot), \boldsymbol{\omega} \rangle | \boldsymbol{\omega} \in \mathbb{R}^d\}$.

**Proposition 2.** *In case of continuous state and action spaces and for $\boldsymbol{\omega} \in \mathbb{R}^d$, the PBO is*

$$\Lambda^*(\boldsymbol{\omega}) = \boldsymbol{\theta} + \gamma \int_\mathcal{S} \max_{a' \in \mathcal{A}} \langle \boldsymbol{\sigma}(s', a') | \boldsymbol{\omega} \rangle \boldsymbol{\mu}(s') \mathrm{d}s'. \tag{4}$$

The closed-form is again a $\gamma$-contraction mapping under the assumption that the MDP has a latent variable representation.

### 4.1.3 LINEAR QUADRATIC REGULATION

Eventually, we consider the continuous MDPs class of linear quadratic regulator (LQR) with $\mathcal{S} = \mathcal{A} = \mathbb{R}$. The transition model $\mathcal{P}(s, a) = As + Ba$ is deterministic and the reward function $\mathcal{R}(s, a) = Qs^2 + 2Ssa + Ra^2$ is quadratic, where $A$, $B$ $Q$, $S$ and $R$, are constants inherent to the MDP. As commonly done for LQR, we consider an undiscounted setting, thus we do not analyze the contraction property of PBO. We choose to parameterize the action-value functions with a 3-dimensional parameter vector $\mathcal{Q}_\Omega = \{(s, a) \mapsto Gs^2 + 2Isa + Ma^2 | (G, I, M) \in \mathbb{R}^3\}$, for visualization purposes.

**Proposition 3.** *The PBO exists and for any $\boldsymbol{\omega} \in \mathbb{R}^2 \times \mathbb{R}_*^-$ [2] its closed form is*

$$\Lambda^* : \boldsymbol{\omega} = \begin{bmatrix} G \\ I \\ M \end{bmatrix} \mapsto \begin{bmatrix} Q + A^2(G - \frac{I^2}{M}) \\ S + AB(G - \frac{I^2}{M}) \\ R + B^2(G - \frac{I^2}{M}) \end{bmatrix}. \tag{5}$$

A geometrical interpretation of the behavior of this PBO is available in Appendix A.

### 4.2 APPROXIMATING PROJECTED BELLMAN OPERATORS

For generic RL problems that do not fall into the previously described categories, the true PBO is unknown and has to be estimated from samples. We propose to approximate the unknown PBO with a *parameterized PBO* $\Lambda_\phi$ differentiable w.r.t. its parameters $\phi \in \Phi$, enabling the use of gradient descent on a loss function. A first idea would be to learn an operator that minimizes the loss in the definition of PBO (Equation 2). While this approach has the benefit of allowing to sample as many parameters as needed from $\Omega$, it does not leverage the crucial fact that approximating an operator allows to iterate over it as many times as desired. Thus, our key insight is to include multiple iterations of the operator to the loss. The function to minimize becomes

$$\mathcal{L}_{\mathrm{PBO}}(\phi) = \sum_{k=1}^K \left( \Gamma^* Q_{\Lambda_\phi^{k-1}(\boldsymbol{\omega})}(s, a) - Q_{\Lambda_\phi^k(\boldsymbol{\omega})}(s, a) \right)^2, \tag{6}$$

for all $\boldsymbol{\omega} \in \Omega$, $s \in \mathcal{S}$, and $a \in \mathcal{A}$, and where $K$ is an arbitrary number of optimal Bellman operator iterations. An additional idea is to add a term that corresponds to an infinite number of iterations, i.e., the fixed point, when possible. Now, the loss becomes

$$\mathcal{L}_{\mathrm{PBO+}}(\phi) = \mathcal{L}_{\mathrm{PBO}}(\phi) + \underbrace{\left( \Gamma^* Q_{\Lambda_\phi^{\boldsymbol{\omega}}}(s, a) - Q_{\Lambda_\phi^{\boldsymbol{\omega}}}(s, a) \right)^2}_{\text{Fixed point term}}, \tag{7}$$

for all $\boldsymbol{\omega} \in \Omega$, $s \in \mathcal{S}$, and $a \in \mathcal{A}$, where $\Lambda_\phi^{\boldsymbol{\omega}}$ is the fixed point of the parameterized PBO. Note that the addition of the fixed point term is only possible for classes of parameterized PBOs where the fixed point can be computed and the computation is differentiable in the parameters $\phi$.

---

[2] If $\boldsymbol{\omega} \in \mathbb{R} \times \{0\} \times \{0\}$, the formula still holds with the convention $0/0 = 0$.

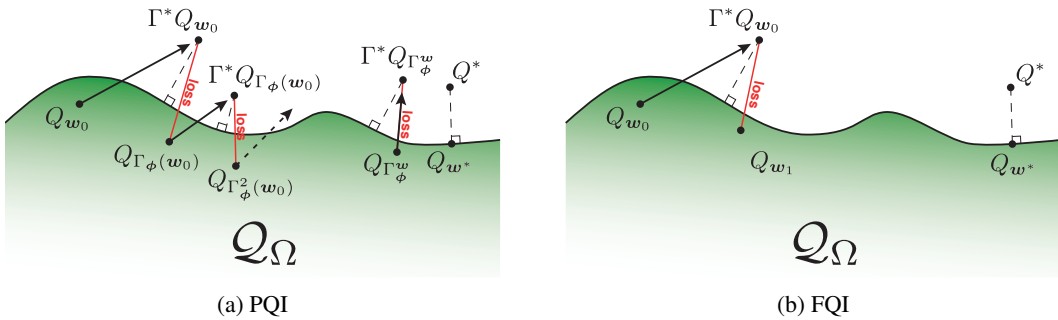

Figure 3: Behavior of PQI and FQI in the space of all value functions for *one* iteration. The dashed lines represent the projections in $\mathcal{Q}_\Omega$.

---

**Algorithm 1** Projected $Q$-Iteration (PQI)

---

1: **Inputs:**
      samples $\mathcal{D} = \{\langle s_j, a_j, r_j, s'_j\rangle\}_{j=1}^J$, parameters $\mathcal{W} = \{\boldsymbol{\omega}_l\}_{l=1}^L$,
      #Bellman iterations $K$, #training steps $I$, #fitting steps $H$,
      operator $\Lambda_\phi$ to learn with initial parameters $\phi = \phi_0$, learning rate $\lambda$
2: **for** $i \in 1 \ldots I$ **do**
3:     # Compute the target $T$
4:     **for** $k \in 1 \ldots K$ **do**
5:         **for** $(\boldsymbol{\omega}_l, (s_j, a_j, r_j, s'_j)) \in \mathcal{W} \times \mathcal{D}$ **do**
6:             $T_l^k(s_j, a_j) = \underbrace{r_j + \gamma \max_{a \in \mathcal{A}} Q_{\Lambda_\phi^{k-1}(\boldsymbol{\omega}_l)}(s'_j, a)}_{\text{optimal Bellman iteration of } Q_{\Lambda_\phi^{k-1}(\boldsymbol{\omega}_l)}}$
7:         **end for**
8:     **end for**
9:     # Minimize the loss starting from the previous parameters
10:     $\phi_i = \phi_{i-1}$
11:     **for** $\_ \in 1 \ldots H$ **do**
12:         **for** $j \in 1 \ldots J$ **do**
13:             $\phi_i \leftarrow \phi_i - \lambda \sum_{l=1}^L \sum_{j=1}^J \sum_{k=1}^K \nabla_\phi \left( T_l^k(s_j, a_j) - Q_{\Lambda_\phi^k(\boldsymbol{\omega}_l)}(s_j, a_j) \right)^2$
14:         **end for**
15:     **end for**
16: **end for**
17: **return** Parameters $\phi_I$ of operator $\Lambda$

---

### 4.3 APPROXIMATE VALUE ITERATION FOR PROJECTED BELLMAN OPERATORS

We devise an approximate value iteration algorithm to learn parameterized PBOs by solving the optimization problems (6) and (7). The general structure of the algorithm closely resembles the one of Fitted $Q$-Iteration (Ernst et al., 2005), alternating one phase in which we compute the targets, and another one where we apply gradient descent to fit the parameters w.r.t. the targets (Algorithm 1). For this reason, we call our algorithm *Projected Q-Iteration* (PQI). When possible, the fixed point of the parameterized PBO is added to the loss. An interesting property of our method is that it enables to build a rich loss by including multiple optimal Bellman iterations at the same time, as opposed to FQI that learns each Bellman iteration sequentially: $Q_i \in \arg\inf_{Q \in \mathcal{Q}} \|\Gamma^* Q_{i-1} - Q\|$. Figure 3 illustrates this advantage by comparing the way PQI and FQI behave in the space of all possible action-value functions. The ability of iterating for an arbitrary number of times enables us to derive theoretical advantages of PQI by leveraging established results in approximation error for AVI.

**Theorem 1.** *(Theorem 3.4 of Farahmand (2011)) Let $K \in \mathbb{N}^*$, $\rho, \nu$ two distribution probabilities over $\mathcal{S} \times \mathcal{A}$. For any sequence $(Q_k)_{k=0}^K \subset B(\mathcal{S} \times \mathcal{A}, R_\gamma)$ where $R_\gamma$ depends on the reward function*

*and the discount factor, we have*

$$||Q^* - Q^{\pi_K}||_{1,\rho} \leq \inf_{r \in [0,1]} F(r; K, \rho, \gamma) \overbrace{\left( \sum_{k=1}^{K} \alpha_k^{2r} \underbrace{||\Gamma^* Q_{k-1} - Q_k||_{2,\nu}^2}_{FQI} \right)^{\frac{1}{2}}}^{PQI} + C_{K,\gamma,R_\gamma}, \quad (8)$$

*where $\alpha_k$ and $C_{K,\gamma,R_\gamma}$ do not depend on the sequence $(Q_k)_{k=0}^K$. $F(r; K, \rho, \gamma)$ relies on the concentrability coefficients of the greedy policies w.r.t. the value functions.*

This theorem shows that the distance between the value function and the optimal value function depends on the approximation errors for all the iterations. The loss of FQI contains only one term of the sum, while the loss of PQI contains the entire sum. In PQI, the terms are summed up with equal weights, which is not the case for the error propagation. Future works could investigate how to weight those terms for increased performance.

## 5 EXPERIMENTS

We empirically analyze our PQI algorithm with different classes of parameterized PBOs w.r.t. FQI (Ernst et al., 2005) and, when possible, the additional baseline least-squares policy iteration (LSPI) (Lagoudakis & Parr, 2003). The fixed point is always added to the loss for the linear PBO, unless stated otherwise. The pseudocodes of FQI and LSPI are provided in Appendix B.

**Experimental details.** Each algorithm receives a fixed dataset of samples $\mathcal{D}$ and has access to the same number of optimal Bellman iterations $K$. We define the *initial parameters* for both FQI and LSPI such that they result in a zero-function. We store the parameters after each iteration of the algorithms; then, we compute the action-value function for each iteration and the value function associated with the greedy policy computed from the action-value function at each iteration. PQI can access the same functions by iterating over the initial parameters for more than $K$ iterations, thanks to our parameterized PBO. When possible, we include the parameters corresponding to an infinite number of iterations, i.e., the fixed point of the parameterized PBO. We learn parameterized PBOs both with linear function approximators (*linear PBO*) and neural networks (*neural PBO*). We also report the performance of the fixed point of the linear PBO (linear PBO fixed point). For problems when PBO is known in closed-form, we include it in the analysis as a reference (*PBO* in the figures). Additional details on the experiments are in Appendix C.

### 5.1 CHAIN-WALK

We initially consider the chain-walk environment in Figure 4, with a chain of length $N = 20$. We parameterize value functions as tables to leverage our theoretical results on finite MDPs (Section 4.1.1). We plot the $\ell_2$-norm between the iterated value functions and the optimal ones in Figure 5a. For all methods, the Bellman operator is iterated for $K = 5$ times during training. The vertical black dotted line denotes the split between the iterations that are learnt and the iteration that can be only reached by parameterized PBO. Notably, we can iterate the learnt PBO for an arbitrary number of times. This property is crucial to nullify the distance between the value

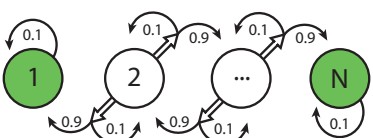

Figure 4: The chain-walk from Munos (2003). Reward is always 0, but 1 in green states.

function computed with the parameterized PBOs and the optimal value function. The superior performance of PQI comes from the fact that both classes of operators are close to the class of the PBO. Since in this setting the PBO perfectly replicates the behavior of the optimal Bellman operator, we can foresee that any algorithm minimizing the approximation error by learning parameters of value functions would perform in the best case as good as the PBO, meaning that their performance would coincide with the one of PBO at iteration 5. For this reason, any algorithm designed as such would perform worse than or equal to FQI in this setting. Note that LSPI is not aiming at reducing the approximation error, this is why the reasoning does not apply to this baseline.

**Analysis of the size of the dataset of parameters.** Ideally, the dataset of parameters should contain parameters close to the initial ones (the parameters with only zeros), others close to optimal

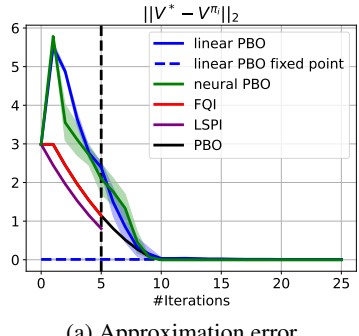 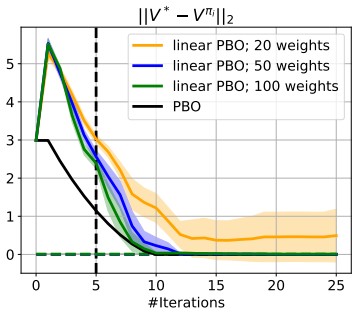

(a) Approximation error.  (b) Ablation on parameters dataset.

Figure 5: $\ell_2$-norm of the optimal value function and the value function induced by the learned greedy policy on chain-walk. Results averaged over 20 seeds with $95\%$ confidence interval.

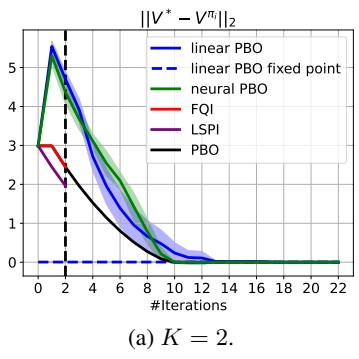 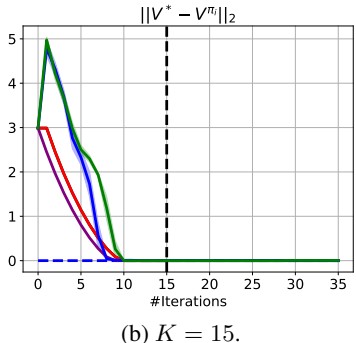

(a) $K = 2$.  (b) $K = 15$.

Figure 6: Analysis on the number of Bellman iterations $K$ on chain-walk.

parameters, and others in between. Since we only assume to know the initial parameters, we take parameters close to them, in the sense that the truncated normal distribution used for data collection is centered. Figure 5b shows the distance to the optimal value function for different number of parameters. Reasonably, the more parameters are given to the algorithm, the better the performance.

**Analysis on the number of Bellman iterations** $K$**.** We additionally analyze the behavior of PBOs for a different number of Bellman iterations than $K = 5$. We consider $K = 2$ and $K = 15$, observing that the PBOs still behave well as shown in Figure 6.

## 5.2  LINEAR QUADRATIC REGULATOR

We consider the LQR setting in Section 4.1.3, leveraging our theoretical analysis by parameterizing value functions accordingly. Again, we consider a neural PBO and a linear PBO. Figure 7a shows a metric proportional to the distance between the optimal value function and the value functions obtained at each iteration. For all methods, we set $K = 2$. As in the previous experiment, the chosen class of operators is close to the class of the optimal PBO. We see that, at iteration 2, the performance of FQI and the PQI are on par, while afterward, PQI keeps improving until the fixed point is reached. Again, we can state that any algorithm that learns parameters of value functions trying to minimize the approximation error, would perform worse than FQI. FQI follows the curve of the PBO, meaning that the training cannot be improved. LSPI performs badly because the first policy for the evaluation step is a uniform policy, and the value function associated with the uniform policy is not a quadratic function.

**Analysis on the number of Bellman iterations** $K$**.** Since the convergence of all the algorithms is fast in term of iterations, we additionally show the performance with 4 Bellman iterations instead of 2. Figures 7b confirms that PBOs perform well for a larger number of Bellman iterations.

**Analysis on the size of the dataset of samples.** We focus on neural PBO. Figure 8a shows the distance to the optimal value function for trainings with a varying number of samples. When the grid becomes thinner, the distances are decreasing. However this decrease is not strong. This is why

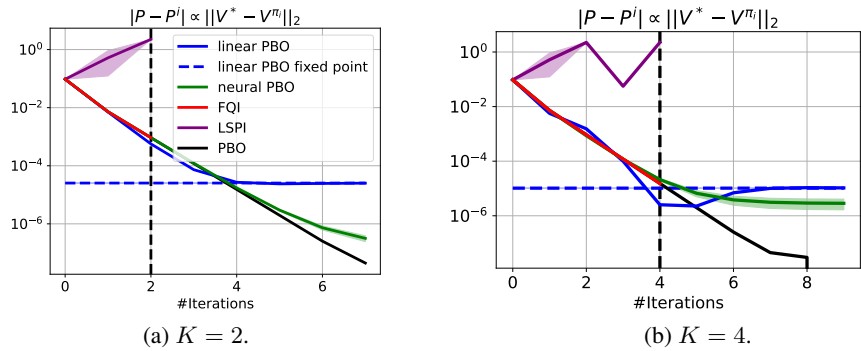

(a) $K = 2$.      (b) $K = 4$.

Figure 7: $\ell_2$-norm of the optimal value function and the value function induced by the learned greedy policy on LQR. Results averaged over 20 seeds with 95% confidence interval.

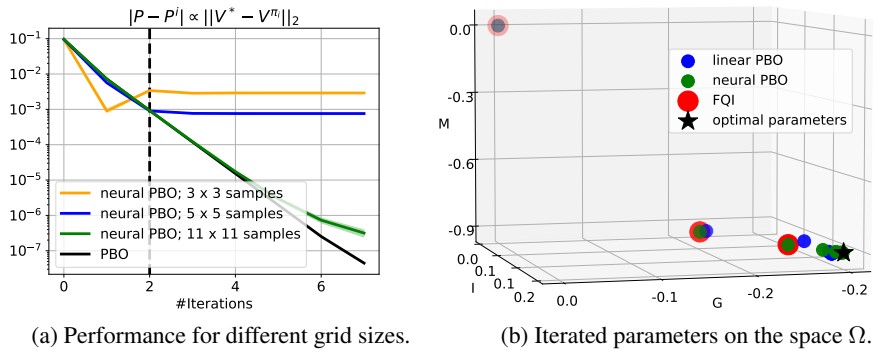

(a) Performance for different grid sizes.      (b) Iterated parameters on the space $\Omega$.

Figure 8: Analysis on number of Bellman iterations and visualization of iterated weights on LQR.

the size of the dataset is not a crucial parameter for learning PBO.

**Visualization.** This setting allow us to plot the parameters of the action-value functions in a 3D space. The parameters corresponding to the iterations of PQI and FQI are shown in Figure 8b. The training is done with $K = 2$. The first parameters for the three algorithms are on $(0, 0, 0)$ and then, the iterations goes towards the optimal parameters on the bottom right of the figure. In this case, being able to iterate after the training clearly makes the difference, as it is clear that PQI is able to get closer to the optimal parameters after the second iteration.

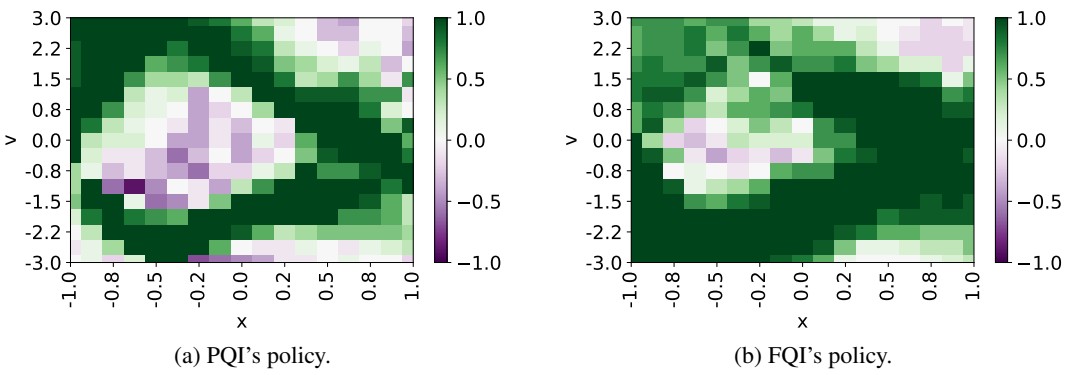

(a) PQI's policy.      (b) FQI's policy.

Figure 9: Policies at iteration 9 on car-on-hill.

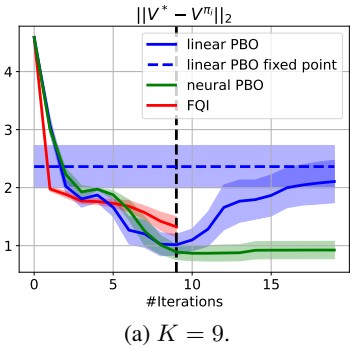 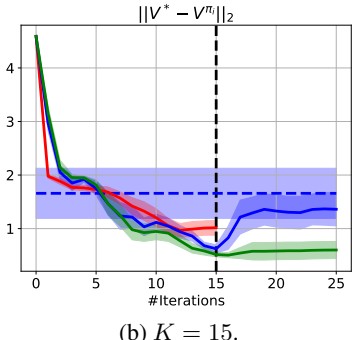

(a) $K = 9$.      (b) $K = 15$.

Figure 10: Weighted $\ell_2$-norm for the value function on car-on-hill (see Appendix C.3). Results averaged over 20 seeds with 95% confidence interval.

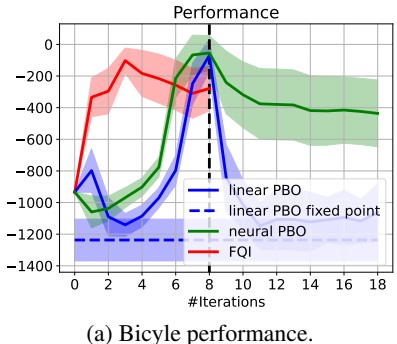 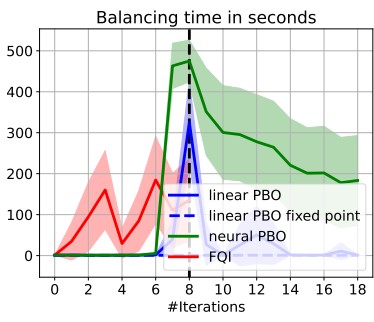

(a) Bicyle performance.      (b) Bicycle balancing time.

Figure 11: Analysis for the Bicycle environment.

### 5.3 NON-LINEAR PROJECTED BELLMAN OPERATORS

We consider a more advanced setting where the value function is approximated by a neural network, thus the space of value functions is not stable by the optimal Bellman operator. We focus on the car-on-hill MDP (Ernst et al., 2005), as implemented in MushroomRL (D'Eramo et al., 2021), and the bicycle MDP in Randlov & Alstrøm (1998). We parameterize value functions with one hidden layer of 30 neurons, and we compare PQI only to FQI, since LSPI is not suitable for this setting. We train a linear PBO and a neural PBO. We set $K = 9$ for car-on-hill and $K = 8$ for bicycle. Figure 9 shows the policies on car-on-hill for FQI and PQI trained with the neural PBO, where PQI is able to obtain a significantly more accurate policy than FQI. Figures 10 and 11a show that PBOs outperform FQI in terms of performance too. Interestingly, iterating further produces a drop in performance for linear PBO, indicating that the class of linear operator is too simple. Remarkably, neural PBO either shows no deterioration (Figure 10a), or a slight deterioration due to opposite performance in different seeds (Figure 11a and Appendix C.4). Further tuning should substantially reduce the number of failing seeds. This experiment evinces the limits of FQI where errors made at any iteration propagates until the last iteration (Theorem 1). Conversely, PQI can change the path to the optimal parameters at each optimization step.

## 6 CONCLUSION

We introduced the original idea of obtaining an approximation of the true Bellman operator, which we call projected Bellman operator (PBO). We defined PBO for different classes of problems, studied their theoretical properties of stability and convergence, and devised a novel approximate value iteration algorithm that provably reduces the error propagation over Bellman iterations. Finally, we have empirically shown the benefit of PBO over related baselines on a variety of problems. We consider this work a seminal study of PBO as a novel paradigm of RL, that can lead to multiple research directions, among which investigating its use in online settings is a first foreseeable avenue.

## 7 REPRODUCIBILITY STATEMENT

We aim to guarantee the reproducibility of our work by providing the proofs of our theoretical results and additional details of the conducted experiments in the appendix. Further, we provide the code for reproducing our results in the supplementary material, and we will make it public upon acceptance.

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

# A PROOFS

*Proof of the closed-form of the PBO for a finite MDP.* We compute the optimal Bellman iteration over a table $Q \in \mathbb{R}^{N \cdot M}$

$$
\Gamma^* Q(s,a) = r(s,a) + \gamma \mathbb{E}_{s' \sim p(\cdot|s,a)} \left[ \max_{a' \in \mathcal{A}} [Q(s',a')] \right]
$$

$$
= r(s,a) + \gamma \sum_{s'} p(s'|s,a) \left[ \max_{a' \in \mathcal{A}} [Q(s',a')] \right]
$$

$$
= \left( R + \gamma P \max_{a' \in \mathcal{A}} Q(\cdot, a') \right)(s,a) \tag{9}
$$

where $P \in \mathbb{R}^{N \cdot M \times N}$ is the transition probability matrix of the environment. From these equations, the operator $Q \mapsto R + \gamma P \max_{a' \in \mathcal{A}} Q(\cdot, a')$, evaluated on the objective function from the definition of PBO 2, yields zero error. This means that we have found the PBO in closed-form. □

*Proof of the closed-form of the PBO for a low-rank MDP.* The proof considers continuous unbounded state action spaces. For $Q_{\boldsymbol{\omega}} \in \mathcal{Q}_{\Omega}$, $\boldsymbol{\omega}$ the vector representing $Q_{\boldsymbol{\omega}}$, $\max_{a \in \mathcal{A}} Q_{\boldsymbol{\omega}}(s,a)$ is well defined (here max might not be attained, it should be interpreted as a supremum). We have $|Q_{\boldsymbol{\omega}}(s,a)| = |\langle \boldsymbol{\sigma}(s,a)|\boldsymbol{\omega} \rangle| \leq ||\boldsymbol{\sigma}(s,a)|| \cdot ||\boldsymbol{\omega}|| \leq ||\boldsymbol{\omega}||$, thus $\max_{a \in \mathcal{A}} Q_{\boldsymbol{\omega}}(s,a) < \infty$. Then, we write the optimal Bellman iteration on the function $Q_{\boldsymbol{\omega}}$ as

$$
\Gamma^* Q_{\boldsymbol{\omega}}(s,a) = r(s,a) + \gamma \mathbb{E}_{s' \sim p(\cdot|s,a)} \left[ \max_{a' \in \mathcal{A}} Q_{\boldsymbol{\omega}}(s',a') \right]
$$

$$
= r(s,a) + \gamma \int_{s'} \max_{a'} \langle \boldsymbol{\sigma}(s',a')|\boldsymbol{\omega} \rangle p(s'|s,a) \mathrm{d}s'
$$

$$
= \langle \boldsymbol{\sigma}(s,a)|\boldsymbol{\theta} \rangle + \gamma \int_{s'} \max_{a'} \langle \boldsymbol{\sigma}(s',a')|\boldsymbol{\omega} \rangle \langle \boldsymbol{\sigma}(s,a)|\boldsymbol{\mu}(s') \rangle \mathrm{d}s' \text{ from the definition of the transition}
$$

probabilities.

$$
= \langle \boldsymbol{\sigma}(s,a)|\boldsymbol{\theta} + \gamma \int_{s'} \max_{a'} \langle \boldsymbol{\sigma}(s',a')|\boldsymbol{\omega} \rangle \boldsymbol{\mu}(s') \mathrm{d}s' \rangle \text{ from the linear of the scalar product.}
$$

This derivation shows that the operator $\boldsymbol{\omega} \mapsto \boldsymbol{\theta} + \gamma \int_{s'} \max_{a'} \langle \boldsymbol{\sigma}(s',a')|\boldsymbol{\omega} \rangle \boldsymbol{\mu}(s') \mathrm{d}s'$ minimizes the objective function presented in the definition of PBO 2 since its yields zero error. This operator is the PBO for a low-rank MDP. □

*Intuition on the architecture of the parameterized PBO for a low-rank MDP.* In the finite case, the PBO is

$$
\Gamma_p^*(\boldsymbol{\omega}) = \boldsymbol{\theta} + \gamma \sum_{s'} \max_{a'} \langle \boldsymbol{\sigma}(s',a')|\boldsymbol{\omega} \rangle \boldsymbol{\mu}(s')
$$

$$
= \boldsymbol{\theta} + \gamma \sum_{s'} \max_{a'} \left( \sum_{i \in \{0,...,d\}} \boldsymbol{\sigma}(s',a')_i \boldsymbol{\omega}_i \right) \boldsymbol{\mu}(s').
$$

We can see that the most suitable architecture is a composition of a linear layer followed by a max-pooling and finally a last linear layer. □

*Proof that the PBO is a $\gamma$-contraction-mapping for a low-rank MDP.* We now assume that the MDP has a latent variable representation. This proof was inspired by the proof of the contraction

property of the Bellman Operator in Bertsekas (2015). Considering $\boldsymbol{\omega}, \boldsymbol{\omega}' \in \mathbb{R}^d$, we have

$$
\begin{aligned}
||\Gamma_p^*(\boldsymbol{\omega}) - \Gamma_p^*(\boldsymbol{\omega}')||_\infty &= \gamma|| \int_{s'} \max_{a'} \langle \boldsymbol{\sigma}(s', a')|(\boldsymbol{\omega} - \boldsymbol{\omega}') \rangle \boldsymbol{\mu}(s') \mathrm{d}s'||_\infty \\
&= \gamma \max_{i \in \{1,...,d\}} \left| \int_{s'} \max_{a'} \langle \boldsymbol{\sigma}(s', a')|(\boldsymbol{\omega} - \boldsymbol{\omega}') \rangle \boldsymbol{\mu}(s')_i \mathrm{d}s' \right| \\
&\leq \gamma \max_{i \in \{1,...,d\}} \int_{s'} \max_{a'} |\langle \boldsymbol{\sigma}(s', a')|(\boldsymbol{\omega} - \boldsymbol{\omega}') \rangle| \, \boldsymbol{\mu}(s')_i \mathrm{d}s' \text{ since } \boldsymbol{\mu}(\cdot) \text{ is positive.} \\
&\leq \gamma \max_{i \in \{1,...,d\}} \int_{s'} \max_{a'} \sum_{j \in \{1,...,d\}} \boldsymbol{\sigma}(s', a')_j |(\boldsymbol{\omega} - \boldsymbol{\omega}')_j| \boldsymbol{\mu}(s')_i \mathrm{d}s' \text{ since } \boldsymbol{\sigma}(\cdot, \cdot) \text{ are positive.} \\
&\leq \gamma \max_{i \in \{1,...,d\}} \int_{s'} \max_{a'} \sum_{j \in \{1,...,d\}} \boldsymbol{\sigma}(s', a')_j \boldsymbol{\mu}(s')_i \mathrm{d}s' \cdot ||\boldsymbol{\omega} - \boldsymbol{\omega}'||_\infty \\
&\leq \gamma \max_{i \in \{1,...,d\}} \int_{s'} \boldsymbol{\mu}(s')_i \mathrm{d}s' \cdot ||\boldsymbol{\omega} - \boldsymbol{\omega}'||_\infty \text{ since } \boldsymbol{\sigma}(\cdot, \cdot) \text{ are probability distributions.} \\
&\leq \gamma ||\boldsymbol{\omega} - \boldsymbol{\omega}'||_\infty \text{ since } \boldsymbol{\mu}(\cdot)_i \text{ is a probability distribution for all } i.
\end{aligned}
$$

$\square$

*Proof of the closed-form of the PBO for LQR MDP.* Let $\boldsymbol{\omega} = (G, I, M) \in \mathbb{R}^2 \times \mathbb{R}_*^-$ [3], the optimal Bellman iteration over $Q_{\boldsymbol{\omega}}$ is $\Gamma^* Q_{\boldsymbol{\omega}}(s, a) = r(s, a) + \max_{a'} Q_{\boldsymbol{\omega}}(s', a')$. Before moving forward, we have to make sure this quantity is properly defined. The function $a' \mapsto Q_{\boldsymbol{\omega}}(s', a') = G \cdot s'^2 + 2I \cdot s'a' + M \cdot a'^2$ is a polynomial of degree 2 in $a'$. By choosing $M < 0$, we ensure that this function has a unique maximum of equation $-I/M \cdot s'$. This is why the PBO is only defined on $\mathbb{R}^2 \times \mathbb{R}_*^- \cup \mathbb{R} \times \{0\} \times \{0\}$. This makes

$$
\max_{a'} Q_{\boldsymbol{\omega}}(s', a') = Q_{\boldsymbol{\omega}}(s', -\frac{I}{M} \cdot s') = G \cdot s'^2 - 2\frac{I^2}{M} \cdot s'^2 + \frac{I^2}{M} \cdot s'^2 = (G - \frac{I^2}{M}) \cdot s'^2.
$$

By putting it in the Bellman equation, we get

$$
\begin{aligned}
\Gamma^* Q_{\boldsymbol{\omega}}(s, a) &= r(s, a) + (G - \frac{I^2}{M}) \cdot s'^2 \\
&= Q \cdot s^2 + 2S \cdot sa + R \cdot a^2 + (G - \frac{I^2}{M}) \cdot s'^2 \\
&= \left( Q + A^2(G - \frac{I^2}{M}) \right) \cdot s^2 + 2 \left( S + AB(G - \frac{I^2}{M}) \right) \cdot sa \\
&\quad + \left( R + B^2(G - \frac{I^2}{M}) \right) \cdot a^2.
\end{aligned}
$$

Then, the following operator is the PBO

$$
\boldsymbol{\omega} \mapsto \begin{bmatrix} Q + A^2(G - \frac{I^2}{M}) \\ S + AB(G - \frac{I^2}{M}) \\ R + B^2(G - \frac{I^2}{M}) \end{bmatrix} \overset{\text{or}}{=} \begin{bmatrix} Q \\ S \\ R \end{bmatrix} + \Lambda(\boldsymbol{\omega}) \begin{bmatrix} A^2 \\ AB \\ B^2 \end{bmatrix},
$$

with $\Lambda(\boldsymbol{\omega}) = G - I^2/M$. $\square$

**Remarks** The PBO can also be understood in a geometrical way. It projects the parameters along a line of direction vector $\begin{bmatrix} A^2 & AB & B^2 \end{bmatrix}^T$ with an offset $\begin{bmatrix} Q & S & R \end{bmatrix}^T$. The iterations correspond to a non-linear transformation of the coefficient in front of the direction vector. This also means that the fixed-point, i.e. the optimal parameters are also on this line.

---

[3] If $\boldsymbol{\omega} \in \mathbb{R} \times \{0\} \times \{0\}$, the result still holds with the convention $0/0 = 0$.

# B  DETAILS OF THE BASELINES

---

**Algorithm 2** FQI

---

1: **Inputs:**
    Dataset of samples $\mathcal{D}$
    #Bellman iterations $K$
    #fitting_steps $H$
    learning rate $\lambda$
2: **Initialize:**
    $\boldsymbol{\omega}_0$ # weight that give the null $Q$ function
3: **for** $i$ in $1 \ldots \mathrm{K}$ **do**
4:     # Compute the target $T$
5:     **for** $(s_j, a_j, r_j, s'_j) \in \mathcal{D}$ **do**
6:         $T(s_j, a_j) = r_j + \gamma \max_a Q_{\boldsymbol{\omega}_{i-1}}(s'_j, a)$
7:     **end for**
8:
9:     # Minimize the loss starting from the previous weights
10:     $\boldsymbol{\omega}_i = \boldsymbol{\omega}_{i-1}$
11:     **for** _ in $1 \ldots H$ **do**
12:         **for** batch $\in \mathcal{D}$ **do**
13:             $\boldsymbol{\omega}_i \leftarrow \boldsymbol{\omega}_i - \lambda \sum_{j \in \text{batch}} \nabla_{\boldsymbol{\omega}} \left(T(s_j, a_j) - Q_{\boldsymbol{\omega}_i}(s_j, a_j)\right)^2$
14:         **end for**
15:     **end for**
16: **end for**
17: **return** $\boldsymbol{\omega}_K$

---

---

**Algorithm 3** LSPI

---

1: **Inputs:**
    Dataset of samples $\mathcal{D}$
    #Bellman iterations $K$
    features $\boldsymbol{\sigma}$
2: **Initialize:**
    Weights $\boldsymbol{\omega}_0$ that give the null $Q$ function
3: **for** $i$ in $1 \ldots \mathrm{K}$ **do**
4:     # Critic step: policy improvement
5:     **for** $(\_, \_, \_, s'_j) \in \mathcal{D}$ **do**
6:         $\pi(s'_j) = \arg\max_a \boldsymbol{\sigma}(s'_j, a)^T \boldsymbol{\omega}_{i-1}$
7:     **end for**
8:
9:     # Actor step: policy evaluation
10:     $\boldsymbol{A} \leftarrow 0$
11:     $\boldsymbol{b} \leftarrow 0$
12:     **for** $(s_j, a_j, r_j, s'_j) \in \mathcal{D}$ **do**
13:         $\boldsymbol{A} \leftarrow \boldsymbol{A} + \boldsymbol{\sigma}(s_j, a_j) \cdot \left(\boldsymbol{\sigma}(s_j, a_j) - \gamma \boldsymbol{\sigma}(s'_j, \pi(s'_j))\right)^T$
14:         $\boldsymbol{b} \leftarrow \boldsymbol{b} + r_j \boldsymbol{\sigma}(s_j, a_j)$
15:     **end for**
16:     $\boldsymbol{\omega}_i = \boldsymbol{A}^{-1} \cdot \boldsymbol{b}$
17: **end for**
18: **return** $\boldsymbol{\omega}_K$

---

Table 1: Summary of all parameters used in the experiments.

| | | Chain-walk | LQR | Car-on-hill | Bicycle |
|---|---|---|---|---|---|
| | horizon | $+\infty$ | $+\infty$ | 100 | 50.000 |
| | $\gamma$ | 0.9 | 1 | 0.95 | 0.99 |
| | #$\mathcal{D}$ | $400^{(*)}$ | $121^{(*)}$ | 5.000 | 70.000 |
| | batch size on $\mathcal{D}$ | 20 | 121 | 500 | 1.000 |
| | K | $5^{(*)}$ | $2^{(*)}$ | $9^{(*)}$ | 8 |
| FQI | fitting step | 400 | 800 | 1.200 | 1.200 |
| | patience | - | - | 7 | 7 |
| | starting learning rate | $10^{-2}$ | $10^{-2}$ | $10^{-3}$ | $5 \times 10^{-3}$ |
| | ending learning rate | $10^{-5}$ | $10^{-5}$ | $10^{-5}$ | $10^{-4}$ |
| PBOs | #$\mathcal{W}$ | $100^{(*)}$ | 1 | 1 | 1 |
| | batch size on $\mathcal{W}$ | 100 | 1 | 1 | 1 |
| | training step | 400 | 3.200 | 4.000 | 500 |
| | fitting step | 4 | 1 | 1 | 15 |
| | starting learning rate | $5 \times 10^{-3}$ | $10^{-2}$ | $10^{-3}$ | $10^{-4}$ |
| | ending learning rate | $10^{-5}$ | $10^{-5}$ | $5 \times 10^{-6}$ | $10^{-7}$ |
| | initial weight std | $5 \times 10^{-4}$ | $5 \times 10^{-4(*)}$ | $5 \times 10^{-4}$ | $5 \times 10^{-9}$ |

$^{(*)}$ Experimental analysis is available for these parameters.

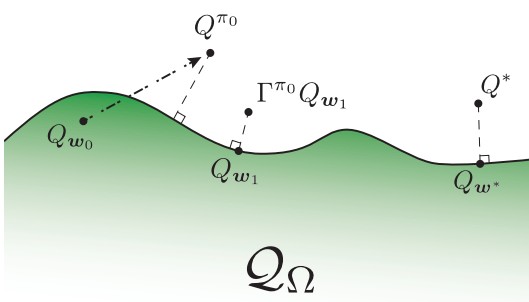

Figure 12: Behavior of LSPI in the space of all action-value functions for *one* iteration. The dotted lines are representing projections in the space of action-value functions.

## C    ADDITIONAL RESULTS AND DETAILS OF THE EMPIRICAL ANALYSIS

We provide details of the experimental setting. Table 1 summarizes the values of all parameters appearing in the experiments. For the additional empirical analysis, only the parameter at stake is changed, all the others remain the ones shown in Table 1. FQI and PQI use an Adam optimizer (Kingma & Ba, 2015) with a linear annealing learning rate. For FQI, the optimizer is reset at each iteration.

### C.1    CHAIN-WALK

We consider all the possible state-action pairs 10 times as the initial dataset of samples $\mathcal{D}$. These 10 repetitions help the algorithms to grasp the randomness of the environment. The set of parameters $\mathcal{W}$ is generated by taking a truncated normal distribution for different seeds. The initial parameters use for the figures is a table composed of zeros. The architecture of the neural PBO is a max-pooling layer followed by a linear layer. This choice has been made so that a perfectly tuned parameterized PBO can be mathematically equal to the PBO.

## C.2 LINEAR QUADRATIC REGULATOR

The dynamics of the system is $\mathcal{P}(s,a) = 0.72s - 0.53a$ and the reward function is $\mathcal{R}(s,a) = -0.13s^2 + 0.26sa - 0.80a^2$. The set $\mathcal{D}$ is collected on a mesh of size 11 by 11 on the state-action space going from $-4$ to $4$ in both directions, which means that samples belong to $[-4,4] \times [-4,4] \subset \mathcal{S} \times \mathcal{A}$. The set $\mathcal{W}$ is composed of only the initial parameters, i.e., $(0,0,0)$. This choice is made because during the optimization, the optimal Bellman operator might project the parameters in an area where the coefficients $M$ are positive which would lead the next optimal Bellman iterations to diverge. For PQI and FQI, we use the optimal Bellman operator, meaning that we should compute the maximum over a value function. We choose to parameterize the value functions in a quadratic way, thus allowing us to compute the maximum in closed-form. However, we assume that the algorithms do not have this knowledge since it is not the case in more general settings. For this reason, we discretize the action space in a set of 200 actions going from $-8$ to $8$. The architecture of the neural PBO starts with a function $\boldsymbol{\omega} \mapsto \boldsymbol{\omega}_1 - \frac{\boldsymbol{\omega}_2^2}{\boldsymbol{\omega}_3}$ and is followed by a linear layer. Once again, this choice has been made to ensure that the neural PBO can reproduce the behavior of the PBO if it is well trained.

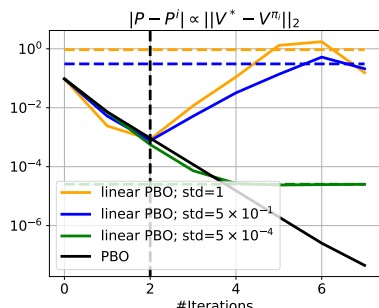

Figure 13: Performance for initial parameters with different standard deviations.

In the following, we explain how to obtain the metric used in Figure 7a. Then, we demonstrate that lowering the standard deviation of the initial parameters of PBO is important.

**Metric.** In this setting, the value functions associated to a greedy policy can be computed in closed form. For a weight $\boldsymbol{\omega} = (G, I, M) \in \mathbb{R}^2 \times \mathbb{R}_*^- \cup \mathbb{R} \times \{0\} \times \{0\}$ such that $A - {}^{BI}/_M \neq 1$, the value function associated to the greedy policy $\pi_{\boldsymbol{\omega}}$ computed from the value function $Q_{\boldsymbol{\omega}}$ has the following form, for any $s \in \mathcal{S}$:

$$V^{\pi_{\boldsymbol{\omega}}}(s) = P_{\boldsymbol{\omega}} \cdot s^2,$$

where $P_{\boldsymbol{\omega}} = \frac{Q - \frac{2SI}{M} + \frac{RI^2}{M^2}}{1 - (A - \frac{BI}{M})^2}$. Note that $G$ does not influence the result, which is normal since the greedy policy is only influenced by the term linked with the actions.

*Proof.* Let $\boldsymbol{\omega} = (G, I, M) \in \mathbb{R}^2 \times \mathbb{R}_*^- \cup \mathbb{R} \times \{0\} \times \{0\}$ such that $A - {}^{BI}/_M \neq 1$, the greedy policy of the value function parameterized by $\boldsymbol{\omega}$ is $\pi^{\boldsymbol{\omega}}(s) = -{}^I/_M \cdot s$ where we take the convention ${}^0/_0 = 0$. To show the result, we show that the function $s \mapsto P_{\boldsymbol{\omega}} \cdot s^2$ satisfies the optimal Bellman equation[4]. We set $P_{\boldsymbol{\omega}} = \frac{Q - \frac{2SI}{M} + \frac{RI^2}{M^2}}{1 - (A - \frac{BI}{M})^2}$, and we have

$\forall s \in \mathcal{S}, P_{\boldsymbol{\omega}} \cdot s^2 = \Gamma^*(x \mapsto P_{\boldsymbol{\omega}} \cdot x^2)(s)$ optimal Bellman equation for the function $x \mapsto P_{\boldsymbol{\omega}} \cdot x^2$.

$\iff \forall s \in \mathcal{S}, P_{\boldsymbol{\omega}} \cdot s^2 = r(s, \pi^{\boldsymbol{\omega}}(s)) + P_{\boldsymbol{\omega}} \cdot s'^2$

$\iff \forall s \in \mathcal{S}, P_{\boldsymbol{\omega}} \cdot s^2 = r(s, -\frac{I}{M}s) + P_{\boldsymbol{\omega}}(As - B\frac{I}{M}s)^2$

$\iff \forall s \in \mathcal{S}, P_{\boldsymbol{\omega}} \cdot s^2 = Qs^2 - 2S\frac{I}{M}s^2 + R\frac{I^2}{M^2}s^2 + P_{\boldsymbol{\omega}}(As - B\frac{I}{M}s)^2$

$\iff P_{\boldsymbol{\omega}} = Q - 2S\frac{I}{M} + R\frac{I^2}{M^2} + P_{\boldsymbol{\omega}}(A - B\frac{I}{M})^2$

$\iff P_{\boldsymbol{\omega}} = \frac{Q - \frac{2SI}{M} + \frac{RI^2}{M^2}}{1 - (A - \frac{BI}{M})^2}.$

Note that if we multiply $s$ to both side of the assumption $A - {}^{BI}/_M \neq 1$, we get $s' \neq s$ where $s'$ is the next state if the greedy policy is followed. This means that we cannot compute the value function of greedy policies that leads to steady states. This is normal since this value function is not defined for all states because the sum of accumulated rewards diverges except in 0 where it is 0. $\square$

---

[4]This only shows the result up to an additive constant, but since $\pi_{\boldsymbol{\omega}}(0) = 0$ then $V^{\pi_{\boldsymbol{\omega}}}(0) = 0$, so this constant is 0.

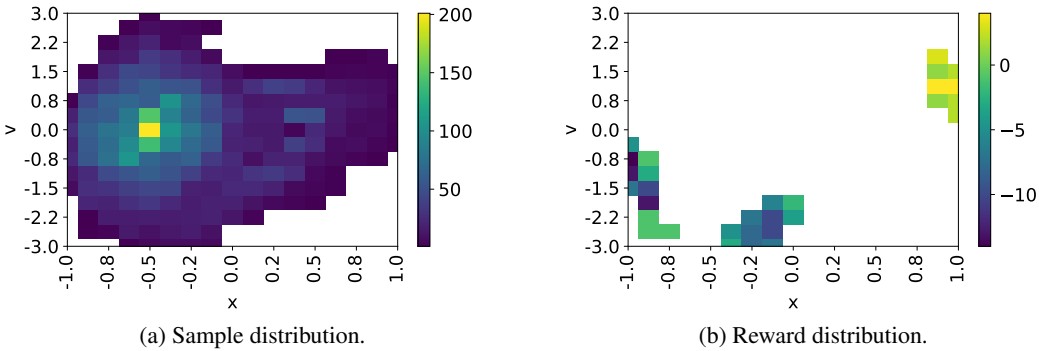

(a) Sample distribution.          (b) Reward distribution.

Figure 14: Composition of the dataset of samples $\mathcal{D}$ on car-on-hill.

We leverage this result to state that the distance between the $P_{\boldsymbol{\omega}}$ of a set of parameters $\boldsymbol{\omega}$ and the solution of the Riccati equation is proportional to the distance between the value function of the greedy policy and the optimal value function.

*Proof.* Let $b \in \mathbb{R}$. We consider the norm-2 over the functions going from $[-b, b]$ to $\mathbb{R}$. For $\boldsymbol{\omega} \in \mathbb{R}^2 \times \mathbb{R}_*^- \cup \mathbb{R} \times \{0\} \times \{0\}$, we have

$$||V^* - V^{\pi_{\boldsymbol{\omega}}}||_2 = \sqrt{\int_{s \in [-b,b]} (V^*(s) - V^{\pi_{\boldsymbol{\omega}}}(s))^2 \mathrm{d}s}$$

$$= |P - P_{\boldsymbol{\omega}}| \sqrt{\int_{s \in [-b,b]} s^4 \mathrm{d}s}$$

$$= |P - P_{\boldsymbol{\omega}}| \cdot \sqrt{2\frac{b^5}{5}}.$$

The norm of the difference of the value functions is proportional to the difference of $P$ coefficients. □

**Analysis on the standard deviation of parameterized PBO's initial parameters.** Figure 13 shows the distance to the optimal value function for linear PBOs that have been initialized with different standard deviations. The linear PBO initialized with smaller standard deviation shows better performance. This behavior can be explained by the following two reasons. First, a linear PBO is very limited in terms of freedom, meaning that if it is initialized in such a way that it goes to the wrong direction, it is very likely that it will stay in the wrong direction during the training; second, by initializing the parameters of the parameterized PBO with low value, the operator is already contractive at the beginning of the training. This helps the parameterized PBO to have a stable behavior when evaluating on iterations that were not seen during training.

## C.3 CAR-ON-HILL

The goal is to bring a car up a hill. The agent chooses between 2 actions: `left` or `right`. The state space is 2-dimensional: position in $[-1, 1]$ and velocity $[-3, 3]$. If the agent succeeds to bring the car up the hill – at position greater than 1 and velocity in between $-3$ and $3$ – then the reward is 1, if the agent exceeds the state space, the reward is $-1$; otherwise, the reward is 0.

As our PQI is an offline algorithm, we need to make sure that all the necessary exploration has been done in the dataset of samples. For that reason, we first consider a uniform sampling policy to collect episodes starting from the lowest point in the map ($[-0.5, 0]$) with an horizon of 100. This sampling process is stopped when $4.500$ samples are gathered. To get more samples with positive reward, we sample new episodes starting from a state located between $[0.1, 1.3]$ and $[0.5, 0.38]$ with a uniform policy as well. In total, $5.000$ samples are collected. The sample and reward distributions over the state space is shown in Figure 14. Inspired by the most suitable architecture for a parameterized

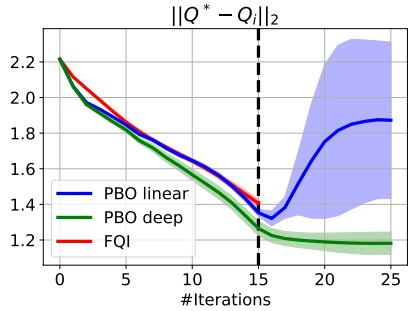 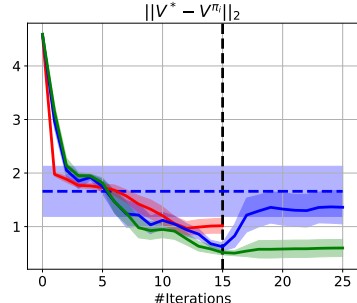

(a) Weighted distance between the optimal action-value function and the learned action-value function for every iteration.

(b) Weighted distance between the optimal value function and the learned value functions for every iteration.

Figure 15: Performance with $K = 15$ on car-on-hill.

PBO, see Appendix A, we set the architecture of the neural PBO with a first block composed of a linear layer followed by a rectified linear unit (ReLU) and another linear layer. Then, the first block is followed by a max-pooling before adding a second block that is identical to the first one. The set of parameters $\mathcal{W}$ is composed of only the initial parameters, because the complexity of the classes of PBOs used in the experiment is not high. The initial parameters are random except for the last layer that is set to zero, ensuring that the associated value function only outputs zeros. To help the training, the value functions are taking actions in $\{-1, 1\}$ instead of the usual $\{0, 1\}$. We also use a low standard deviation for the initial parameters of the PBOs as described in Appendix C.2.

**Metric.** Ernst et al. (2005) measure the performance of the agents by performing roll-outs starting from different states on a grid of size $17 \times 17$. Since the policies, the reward, and the dynamics are deterministic, the discounted reward obtained at the end of the roll-outs corresponds to the value function associated with the policy used during the roll-outs. To compute the optimal value function, they run a brute force algorithm that tries every possible sequence of actions until it finds the shortest way to go up hill. In our case, the agents do not have all the transitions in $\mathcal{D}$ corresponding to every starting states of the grid they are using, as shown in Figure 14a. This is why we apply a mask on the grid to weight the starting states by the number of time a state has been visited in the dataset.

**Analysis on the number of Bellman iterations $K$.** Figure 15 shows that PQI outperforms FQI at iteration 15, for $K = 15$. Similarly to the case $K = 9$, the linear PBO is too simple to behave well after the training iterations. Once again, the way the loss is shaped allows PQI to outperform FQI.

## C.4 BICYCLE

We consider the bicycle problem, as described in Randlov & Alstrøm (1998). The state space is composed of 4 dimensions: $(\omega, \dot{\omega}, \theta, \dot{\theta})$ where $\omega$ is the angle between the floor and the bike, and $\theta$ is the angle between the handle bar and the perpendicular axis to the bike. The goal is to ride a bicycle for 500 seconds (50.000 steps). The agent can apply a torque $T \in \{-2, 0, 2\}$ to the handle bar to make it rotate. The agent can also move its center of gravity in the direction $d \in \{-0.02, 0, 0.02\}$ perpendicular to the bike. As in Lagoudakis & Parr (2003), the agent chooses between applying a torque or moving its center of gravity, resulting in 5 actions instead of 9. Usually, a uniform noise in the interval $[-0.02, 0.02]$ is added to $d$. For the purpose of this work, we reduce the number of samples by making the magnitude of the noise 10 times smaller. A reward of $-1$ is given when the bike falls down, i.e., $|\omega| > 12°$. We also use reward shaping to have more informative samples, and we add a reward proportional to the change in $\omega$, i.e., $10^4(|\omega_t| - |\omega_{t+1}|)$, as in Lagoudakis & Parr (2003). The dataset of samples is composed of 3.500 episodes starting from a position close to $(0, 0, 0, 0)$ and cut after 20 steps (Lagoudakis & Parr, 2003). The architecture of the neural PBO is the same as the one used in the Car-On-Hill experiment. To evaluate the value functions, we simulate 10 trajectories. We observed only two kinds of outcomes in our experiments: either the policies deteriorate the performance quickly, or they keep it stable, as shown in Figure 11b. This also explains the big confidence interval for deep PBO in Figure 11a.

