# OpenReview forum: "Parameterized projected Bellman operator"
_ICLR.cc/2023/Conference — Submitted to ICLR 2023_

### Official Review · Reviewer_bapX · 2022-10-25

**Confidence:** 3
**Clarity, Quality, Novelty And Reproducibility:** Please refer to the weakness section.
**Correctness:** 3
**Technical Novelty And Significance:** 3
**Empirical Novelty And Significance:** 2
**Recommendation:** 5

**Strength And Weaknesses:**

Strength:

  The paper discusses an essential problem in the literature.

  Also, the author provides a nice solution to tackle the problem, but the analysis and intuition part is very poorly represented in this paper.

Weaknesses:

" first to deal with the original problem of operator learning in RL."
I don't think this statement is correct. Please refer [R1] and BBO(Fellows et al., 2021) from the paper.

The author should compare the results of other Bellman operators like the Bayesian Bellman operation. It would be nice to observe the performance of the PBO with other Bellman operators' methods.

It is not clear in the paper how the parameter approximation is formulated in function space (Operator learning space). Which function approximation method is used in PBO? How does it work in the operation space? The author should add a paragraph on function approximation in the literature.

The author should show the parameter distribution of V before the approximation and after the approximation.

How does the quality of the sample improve in PBO techniques? The author should show results both qualitatively and quantitatively.

How about the direction of the update by the current sample? The decision is taken by a single instance or batch of samples.

The author should report ablation analysis on various parameter approximation methods.

The experiment section is not represented properly. The author should report stability results to compare with another method. Also should report the approximation error results.

Reference:
R1. Song, Zhao, Ron Parr, and Lawrence Carin. "Revisiting the softmax bellman operator: New benefits and new perspective." In International conference on machine learning, pp. 5916-5925. PMLR, 2019.

**Summary Of The Paper:**

The author has discussed the problem of transition samples in the bellman operator, which determine the operator's behavior. The author has proposed a novel idea of approximation of the Bellman operator known as the Projected Bellman Operator (PBO). Given the value function's parameter, PBO outputs the new value function's parameter. PBO is a parametric operator on the parameter space of the given value function. Finally, the author has analyzed PBO in terms of stability, convergence, and approximation error on different problems. The author claims that the PBO method overcomes the classical methods' limitations.

**Summary Of The Review:**

The overall writing quality is ok, and the proposed method is simple and beneficial. However, the experiment comparison lacks justification, and the technical contribution is plain.

---

> ### Author Response · Authors · 2022-11-13
> **Response to Reviewer bapX**
>
> We thank the reviewer for the time spent on the submission and the comments that have helped us craft a better version of our work. We now tackle the raised questions / comments one by one.
>
> > [...] the analysis and intuition part is very poorly represented in this paper.
>
> We agree that the core idea of our work is not straightforward to grasp in the first submission. In the revised submission, the organization of the first 5 sections has been rearranged to present the analysis and the intuition parts in a much clearer way in our opinion.
>
> > first to deal with the original problem of operator learning in RL." I don't think this statement is correct. Please refer [R1] and BBO(Fellows et al., 2021) from the paper.
>
> We argue that in [R1] the proposed Bellman operators are not learned during training. Those operators are derived from mathematical analyses. Similarly, the Bayesian Bellman operator is not learned in closed form. This operator is parameterized on top of the usual empirical Bellman operator. In BBO (Fellows et al., 2021), only the parameters of the $Q$ functions are learned.
>
> > The author should compare the results of other Bellman operators like the Bayesian Bellman operation. It would be nice to observe the performance of the PBO with other Bellman operators' methods.
>
> In this work we focus on the optimal Bellman operator. We define a new operator based on the optimal Bellman operator that we call the projected Bellman operator and we approximate it with a neural network. As stated in the related works section, “We point out that our PBO can be seamlessly applied on an arbitrary variant of the standard Bellman operator”. This means that one can define the projected Bayesian Bellman operator and try to approximate it with a neural network. For that reason, we made the choice of not comparing our method to other Bellman operators since it is not the focus of the work.
>
> > It is not clear in the paper how the parameter approximation is formulated in function space (Operator learning space). Which function approximation method is used in PBO? How does it work in the operation space? The author should add a paragraph on function approximation in the literature.
>
> The closed form solution of the PBOs are presented in Section 4 of the revised version. The class of operators used in the experiments are either linear operator or non-linear operator learned with neural networks. We hope that the restructure of Section 4 helps to clarify the doubts.
>
> >The author should show the parameter distribution of V before the approximation and after the approximation.
>
> In this work, we are only focusing on $Q$ functions. The value functions mentioned in the figures are corresponding to the greedy policy of the obtained $Q$ functions. In the revised submission, we show the parameters of the $Q$ functions for all the iterations of the LQR environment. Furthermore, we show the policies obtained by PQI and FQI on the last iteration of the Car-On-Hill environment.
>
> > How does the quality of the sample improve in PBO techniques? The author should show results both qualitatively and quantitatively.
>
> We now provide an analysis of the size of the offline dataset of samples given as input to PQI for two environments: Chain-Walk and LQR. The size of the offline dataset directly impacts the quality of the samples, especially for the Chain-Walk environment. The Chain-Walk environment is a very stochastic environment, reducing the number of samples means reducing the quality of the samples.
>
> > How about the direction of the update by the current sample? The decision is taken by a single instance or batch of samples
>
> In Table 1 in the Appendix, the batch size for every environment is specified. It is not reported in the submission but we found that having an excessively small or big affects the performances of the algorithms.
>
> > The author should report ablation analysis on various parameter approximation methods.
> The experiment section is not represented properly. The author should report stability results to compare with another method. Also should report the approximation error results.
>
> Please find 7 analyses on 4 different parameters and some analyses on the stability of PQI in the empirical section.

---

### Official Review · Reviewer_emHW · 2022-10-25

**Confidence:** 4
**Correctness:** 3
**Technical Novelty And Significance:** 3
**Empirical Novelty And Significance:** 3
**Recommendation:** 6

**Clarity, Quality, Novelty And Reproducibility:**

**Clarity:**
The paper is very clearly written, and the included Figures are clear and helpful for understanding/summarizing concepts.

**Quality:**
The only concerns I have about the paper's quality are: ~~(1) no connection is made between the proposed idea and model-based RL (including related works),~~ (2) the lack of a comprehensive discussion of the limitations of the proposed algorithm, and (3) the experiments seem more like demonstrations than rigorous comparisons of each algorithm due to not checking other values of parameters used in the experiments.

**Novelty:**
~~I'm unsure of how exactly the idea of PBOs differs from existing algorithms that learn parametric models of the transition dynamics and reward function for use in learning a value function (e.g., the Dyna algorithm). However,~~ the idea seems to be explored in a different way, resulting in different theoretical analyses and yielding a different final algorithm.

**Reproducibility:**
Enough details are given in the paper and appendices that the results could likely be reproduced without much difficulty. Thank you, authors!

**Strength And Weaknesses:**

**Strengths:**
1. Very clearly written. Thank you!
1. Examples are given when explaining concepts, which is very helpful.
1. Plots are well-done, large enough to read, and include 95% confidence intervals.

**Potential Weaknesses/Questions:**
1. The main advantage of learning a PBO as stated in the introduction is that it can be applied an arbitrary number of times without using samples. Applying it repeatedly in this way would require the PBO to be extremely accurate, as any error would be propagated and magnified with each application. However, parametric models are usually biased by the initial values of the model parameters, and therefore are likely to be inaccurate while the parameters are optimized. In addition, asymptotic accuracy is limited by how well the chosen function class can represent the true Bellman operator. For these reasons, it seems like it would be difficult to actually take advantage of the main strength of a PBO in practice, especially on larger/harder problems where the function approximator is not powerful enough, where it's unclear what parameterization is appropriate, or whenever a huge batch of data is not available (such as the usual online setting).
1. When the PBO parameterization is not suitable, the algorithm can fail completely. This seems like a problem, because knowledge of which parameterization is sufficient for a given environment is not available in general.
1. ~~How is learning a PBO different from learning a parametric model of the transition and reward functions in model-based RL (e.g. Dyna)? This connection with model-based RL should definitely be discussed in the Related Works section.~~ The authors' comments and the updated version of the paper made it clearer to me that there is no actual model of the environment being used to generate samples in the update.
1. Experiments are on simple problems with powerful function approximation. How does the method perform on harder problems with weaker function approximation?
1. The paper does not sufficiently discuss weaknesses of the proposed method.
1. The paper only presents the parameter settings used in the experiments, but does not mention how these parameters were chosen, what other values were tried, or how different values of parameters change the resulting performance of each method (although this is done for a subset of the parameters). Hence, the experiments seem more like demonstrations than rigorous comparisons between algorithms.
1. PBO seems to only be specified for the offline batch setting, which is at odds with the usual online sequential RL setting. How could it be used in the online setting?
1. Learning a PBO requires transition samples, so why would using transition samples to learn a PBO and then using the PBO to learn a parametric value function be better than using the transition samples to learn a value function directly?
1. ~~What is the vertical dashed black line in the plots?~~ The authors addressed this in their response.
1. Consider using a colour scheme for the plots that is more friendly to people with colourblindness. Alternatively, the legend could be replaced with labels that point to each line, so that the ability to discern colour is not actually necessary to tell which algorithm corresponds to which line.
1. The name "projected Bellman operator" is very similar to the "projected Bellman error" (i.e., the Bellman error with a projection operator) used in gradient TD algorithms, which was a bit confusing. This is an extremely minor point, but it might be good to explicitly clarify the difference.

**References:**
- Sutton, R. S., Szepesvári, C., Geramifard, A., & Bowling, M. (2008, July). Dyna-style planning with linear function approximation and prioritized sweeping. In Proceedings of the Twenty-Fourth Conference on Uncertainty in Artificial Intelligence (pp. 528-536).

**Summary Of The Paper:**

The paper proposes learning a parametric model of the Bellman operator, and using this Projected Bellman Operator (PBO) in place of the true Bellman operator to optimize a learned value function. The paper gives examples of PBOs for several settings, and derives an approximate value iteration algorithm based on PBOs. Experiments on several environments are conducted.

**Summary Of The Review:**

~~My initial recommendation is to reject the paper due to some of the weaknesses mentioned above. However, I actually quite like the paper and the idea of projected Bellman operators, and feel that the paper is not far from clearing the bar for acceptance; if the authors respond satisfactorily to my concerns, I would recommend acceptance.~~

The authors have responded satisfactorily to most of my concerns, although I would still like to see a discussion of the weaknesses/limitations of the proposed algorithm. For that reason I'm comfortable recommending the paper be accepted.

---

> ### Author Response · Authors · 2022-11-13
> **Response to Reviewer emHW**
>
> We thank the reviewer for the extensive feedback. It seems that this work has raised doubts we are happy to clear.
>
> > [...] However, parametric models are usually biased by the initial values of the model parameters, and therefore are likely to be inaccurate while the parameters are optimized. [...]
>
> This remark is completely true. However, it can also be applied to any other algorithm aiming at learning some parameters and is not proper to our method. Nevertheless, we still provide an experiment in which we make the initial standard deviation (in the appendix) of the parameters of the PBO vary and study its different behaviors.
>
> > [...] In addition, asymptotic accuracy is limited by how well the chosen function class can represent the true Bellman operator. [...]
>
> We agree with the reviewer. When it is the case, the major advantage of our method is that it takes into account a larger bound of the propagation error as shown in Theorem 1. This leads our PBO to outperform the baselines on the training iterations on the environment Car-On-Hill and Bicycle.
>
> > When the PBO parameterization is not suitable, the algorithm can fail completely. This seems like a problem, because knowledge of which parameterization is sufficient for a given environment is not available in general.
>
> It is true, however in practice, a PBO can still outperform the baselines thanks to its richer loss. This behavior has been observed in the two environments on which we do not have prior knowledge, namely the Car-On-Hill and Bicycle environments. Furthermore, this behavior can be understood through the error propagation as shown in Theorem 1.
>
> > How is learning a PBO different from learning a parametric model of the transition and reward functions in model-based RL (e.g. Dyna)? This connection with model-based RL should definitely be discussed in the Related Works section.
>
> PBO is a model-free method that does not rely on learning a transition model or a reward function. Nevertheless, extending to model-based is an interesting direction for future work.
>
> > Experiments are on simple problems with powerful function approximation. How does the method perform on harder problems with weaker function approximation?
>
> We agree that it would be interesting to investigate other environments with more complex environments. However, more complex environments would require an online setting or at least to change the set of initial parameters given to PQI during the training. This would lead to another algorithm that is outside of the scope of this work in our opinion. It is definitely a direction that we would like to pursue in future works.
>
> > The paper only presents the parameter settings used in the experiments, but does not mention how these parameters were chosen, what other values were tried, or how different values of parameters change the resulting performance of each method (although this is done for a subset of the parameters). Hence, the experiments seem more like demonstrations than rigorous comparisons between algorithms.
>
> We agree with the reviewer that the empirical analysis of the original submission was suboptimal. Therefore, we moved several empirical results from the appendix to the main paper.
>
> > PBO seems to only be specified for the offline batch setting, which is at odds with the usual online sequential RL setting. How could it be used in the online setting?
>
> First, we consider the online setting to be out of the scope of this work, since our focus is currently on offline RL. The question is still interesting for future works. The only aspect that has to be decided is the sampling policy. We believe that the best way of sampling new samples would be to use an $\varepsilon$-greedy policy over the $Q$ function iterated $K$ times with the learned PBO, if $K$ is the number of iterations learned during training.
>
> > Learning a PBO requires transition samples, so why would using transition samples to learn a PBO and then using the PBO to learn a parametric value function be better than using the transition samples to learn a value function directly?
>
> Learning a PBO gives the possibility to go further in the iterations. This property allows us to build a richer loss that leads the PBO to move the iterated $Q$ functions closer to the optimal value function than the sequential approach of approximate value iteration.
>
> > What is the vertical dashed black line in the plots?
>
> It represents the split between the learnt iterations (i.e. the iterations that have been seen during training) and the iterations that can be computed with the learnt PBO after the training. This has been specified in the revised submission. Thank you for pointing it out.
>
> > Consider using a color scheme for the plots that is more friendly to people with color blindness. Alternatively, the legend could be replaced with labels that point to each line, so that the ability to discern color is not actually necessary to tell which algorithm corresponds to which line.

---

> ### Author Response · Authors · 2022-11-22
> **Second response to Reviewer emHW**
>
> We thank the reviewer for the effort in reading our reply and the revised version of our work, and for increasing the score. We are happy to address the main limitations of our work in the following, and we would be willing to include them in the final version of the paper.
>
> - **PBO in high-dimensional MDPs:** the prominent limitation of our formulation of the projected Bellman operator (PBO) is its ability to scale to high-dimensional problems. We have shown how to use PBO with linear function approximation and neural networks, but we decided to postpone the analysis of PBO for deep neural networks to future work. Notably, our experiments using neural networks for learning PBOs (Section 5.3) already give us the hint that using PBO in high-dimensional problems is certainly possible, but it will require the use (or introduction) of technical advances to handle the large number of action-value function parameters, e.g., parameter sharing with convolutions, model compression, and so on. Thus, we firmly believe that our formulation of PBO does not hinder scalability in general and that this issue could be surely addressed with the aforementioned technicalities.
> - **Hyperparameters:** learning an operator like PBO comes with the risk of obtaining divergent behavior of the action-value function update, especially in the early phases of learning. However, we showed that this risk can be controlled by lowering the standard deviation of the initial parameters of a neural PBO (see Figure 13). Doing so allows us to start the training with an operator that is already a contraction-mapping, thus limiting the risk of divergence.
> - **One-step application of Bellman operator:** this is not a limitation of PBO, but rather an interesting extension that we postpone to future work. Our current formulation of PBO approximates a *single* step of the Bellman operator. It is interesting to study how to enable PBO to approximate *multiple* iterations at once. This would require an additional theoretical analysis that go beyond the scope of this work, but it is definitely worth studying in the future.

---

### Official Review · Reviewer_Kjbj · 2022-10-30

**Confidence:** 5
**Correctness:** 3
**Technical Novelty And Significance:** 1
**Empirical Novelty And Significance:** 2
**Recommendation:** 3

**Clarity, Quality, Novelty And Reproducibility:**

Please see the detailed comments in the previous section.



**Strength And Weaknesses:**

Strength

The paper considers a new paradigm for value-based RL, which could be potentially interesting for both theoretical and practical applications. Other than some notational ambiguity, the paper is well written. I also like that the authors give some specific examples of the proposed general operator.

Weakness

The major weakness of this paper is that the significance of the contributions is not clear to me.

Since the main contribution is PBO, the key to decide if the contribution is significant enough is to understand if this new operator can bring some new findings from either empirical or theoretical perspective. However this is not clear to me as explained in details below. Please correct me if I missed anything important in the rebuttal.


1. The so called PBO, defined in Equations 3 and 4, is defined as the solution of solving a single step of fitted value iteration for a general parameterized function class. But I believe this operator has already been considered in the literature. For example, [1] defines it in a similar way, and analyze FVI with this operator in a batch setting.

2. The authors claim that “our PBO can approximate repeated applications of the true Bellman operator at once, as opposed to the sequential nature of the standard Bellman operator”. This is what motivates the learning objective 13. However, to avoid a sequential implementation when computing $T_\phi^k$, doesn’t it need the assumption that the operator is linear, or at least can be written as a closed form? I believe this is definitely not true for general non-linear function approximation.

3. The authors consider three cases as examples of PBO and gives convergence analysis for all three examples. However, it seems that all these results are already known? For example, a more general case of tabular MDP (Section 4.2) is under the assumption of linear function approximation (as one-hot feature is linear), in which case similar results have been provided in [2].

4. Some notations might not be well defined. For example, What is F in Theorem 1? What is $\Pi$ in Equation 6? Equation 13: there is no (s,a).



[1] Chen, J. and Jiang, N., 2019, May. Information-theoretic considerations in batch reinforcement learning. In International Conference on Machine Learning (pp. 1042-1051). PMLR.

[2] Parr, R., Li, L., Taylor, G., Painter-Wakefield, C. and Littman, M.L., 2008, July. An analysis of linear models, linear value-function approximation, and feature selection for reinforcement learning. In Proceedings of the 25th international conference on Machine learning (pp. 752-759).


**Summary Of The Paper:**

This paper proposes the projected Bellman operator (PBO), an operator on the parameter space of the value function, for value-based reinforcement learning. The paper argues that PBO can approximate repeated applications of Bellman operator at once. To show the advantage of this, the paper considers three example of applying PBO, including tabular MDP, low-rank MDP and LQR, where the convergence and approximation error of PBO are provided. Finally, the paper also provides empirical studies in simple synthetic environments.


**Summary Of The Review:**

I recommend rejecting this paper as the contribution of the proposed method is not clear to me.

---

> ### Author Response · Authors · 2022-11-13
> **Response to Reviewer Kjbj**
>
> We thank the reviewer for the insightful comments and the time spent reviewing the paper. We noticed that, in our opinion, most comments are actually due to misunderstandings about our work. For this reason, we did our best to clarify the contributions of our work in the submitted revised version. We thank the reviewer for bringing up this lack of clarity, we believe that it has improved the quality of this work significantly. We would really appreciate it if the reviewer could spend time checking the revised submission. We now discuss the points one by one.
>
> > The major weakness of this paper is that the significance of the contributions is not clear to me.
>
> We agree with the reviewer. In the introduction of the revised submission, the contributions of the work are now clearly explained.
>
> > The so-called PBO, defined in Equations 3 and 4, is defined as the solution of solving a single step of fitted value iteration for a general parameterized function class. But I believe this operator has already been considered in the literature. For example, [1] defines it in a similar way, and analyzes FVI with this operator in a batch setting.
>
> We believe that there has been a misunderstanding. On the one hand, in the cited paper, the minimization is done on the space of parameters of the $Q$ functions (the parameters are noted $\omega$ in the submission). On the other hand, in our work, the minimization is done on the space of operators over the \textit{parameters} of the $Q$ functions (the operators are noted $\Lambda_\phi$ in the submission). Due to this difference, in the cited papers, minimizing the objective function results in solving a single step of fitted value iteration whereas in our work, minimizing the objective function results in solving several steps of fitted value iteration.
>
> > The authors claim that “our PBO can approximate repeated applications of the true Bellman operator at once, as opposed to the sequential nature of the standard Bellman operator”. This is what motivates the learning objective 13. However, to avoid a sequential implementation when computing $T_{\phi}^k$, doesn’t it need the assumption that the operator is linear, or at least can be written as a closed form? I believe this is definitely not true for general nonlinear function approximation.
>
> First of all, in order to avoid any possible confusion we rephrased the sentence: “our PBO can approximate repeated applications of the true Bellman operator at once, as opposed to the sequential nature of the standard Bellman operator” of the abstract. To make it clear: PBO approximates repeated applications of the true Bellman operator by iterating over the parameter space, but one iteration of a PBO approximates only one application of the true Bellman operator.
> Then, a PBO is learned to approximate the behavior of the optimal Bellman operator on the space of parameters of the $Q$ functions. The operator that the PBO has to learn does not have to be known in closed-form. If the neural network representing the PBO is large enough, it can approximate any operator. Nonetheless, we showed (for finite, low-rank, and LQR MDPs) that not only the operator that has to be learned is known in closed-form, but it also has a simple architecture, for example max-pooling + linear layer for finite MDPs.
>
> > The authors consider three cases as examples of PBO and give convergence analysis for all three examples. However, it seems that all these results are already known? For example, a more general case of tabular MDP (Section 4.2) is under the assumption of linear function approximation (as one-hot feature is linear), in which case similar results have been provided in [2].
>
> We think that the confusion that has been made in point 1. has led the reviewer to misunderstand section 4. In section 4, we first define the projected Bellman operator (PBO) and then compute its closed form for three different classes of MDPs, namely: Finite MDP, Low-Rank MDP, and LQR MDP. In the paper cited by the reviewer, the main focus is on value functions and not on the projected Bellman operator.
>
> > Some notations might not be well defined. For example, What is $F$ in Theorem 1? What is $\Pi$ in Equation 6? Equation 13: there is no $(s,a)$.
>
> Thank you for spotting these mistakes. The revised submission includes the corrections. Concerning $F$ in Theorem 1, we choose not to develop it further since it does not play a major role in the understanding of the error propagation bound.

---

### Official Review · Reviewer_3oQg · 2022-11-03

**Confidence:** 3
**Correctness:** 4
**Technical Novelty And Significance:** 3
**Empirical Novelty And Significance:** Not applicable
**Recommendation:** 6

**Clarity, Quality, Novelty And Reproducibility:**

Clarity and Reproducibility: The paper is well organized and easy to follow. However, the authors may want to provide a detailed discussion on advantages of PBO over the standard Bellman operator.

Quality and Novelty: The idea of this paper is novel to me. However, the experiments need to be improved.

**Strength And Weaknesses:**

Strengths:

1.	To the best of my knowledge, the idea of obtaining a parametric operator on the parameter space of a given value function is novel.

2.	The proposed method is theoretically sound. The authors show how to derive different PBOs according to the class of problems at hand.

3.	The paper is well organized and easy to follow.

Weaknesses:

1.	The experiments can be improved. It would be more convincing if the authors could evaluate PBO on more challenging continuous control tasks, such as Mujoco [1].

2.	The authors may want to provide a detailed discussion on advantages of PBO over the standard Bellman operator.

[1] Todorov et.al "Mujoco: A physics engine for model-based control." international conference on intelligent robots and systems. IEEE, 2012.

**Summary Of The Paper:**

This paper proposes to obtain an approximation of the standard Bellman operator, called projected Bellman operator (PBO). Specifically, PBO is a parametric operator on the parameter space of a given value function. Based on the idea of PBO, the authors develop a novel algorithm for value estimation. Experiments demonstrate the effectiveness of the proposed method.

**Summary Of The Review:**

To the best of my knowledge, the idea of obtaining a parametric operator on the parameter space of a given value function is novel. The paper is well organized and easy to follow. However, the experiments can be improved (please see above for details).

---

> ### Author Response · Authors · 2022-11-13
> **Response to Reviewer 3oQg**
>
> We thank the reviewer for the useful suggestions.
>
> > The experiments can be improved. It would be more convincing if the authors could evaluate PBO on more challenging continuous control tasks, such as Mujoco [1].
>
> We agree that it would be interesting to investigate other environments with more complex dynamics / settings. However, more complex environments would require an online setting or at least to change the set of initial parameters given to PQI during the training. This would lead to another algorithm that, although interesting, deserves an accurate extended analysis that is outside of the scope of this work, and that can be postponed to future work.
>
> > The authors may want to provide a detailed discussion on advantages of PBO over the standard Bellman operator.
>
> We improved the description of our method to clarify its advantages over regular AVI. We report here an extract of the improved description:
>
> - Limitations of AVI:
>
> "Intuitively, the dependence of value iteration on the samples has an impact on the efficiency of the algorithms and on the quality of the obtained estimated value function, which becomes accentuated when solving continuous problems that require value-based methods with function approximation, e.g., approximate value iteration (AVI). Moreover, in AVI approaches, costly function approximation steps are needed to project the output of the Bellman operator back to the considered action-value functional space."
>
> - Advantages of PBO:
>
> "The crucial advantages of our approach are twofold: (i) the output of PBO always belongs to the considered action-value functional space, avoiding, therefore, the costly projection step typical when using the Bellman operator, and (ii) once learned, PBO is applicable for an arbitrary number of iterations without using further samples"

---

### Author Response · Authors · 2022-11-13
**Response to all Reviewers**

We want to thank all the reviewers for their insightful comments, which highlighted the issues of the first version of our submission, and their suggestions to improve the paper. In the revised version, we addressed the common concerns of the reviewers, by making the following changes.

- The abstract and the introduction have been slightly updated to clarify the contribution better;
- Section 4 has been significantly restructured and rephrased:
    - We eliminated the distinction between $\pi$-PBO and optimal PBO, considering only optimal PBO simply calling it PBO;
    - We removed the discussion about the stability of the PBO, which had no practical implication in our work;
    - We present the closed forms of PBO for each problem in a more compact way;
- We eliminate the distinction between the different classes of non-linear PBO, calling them “parameterized PBO”. The practical implementation is called "linear PBO" in the case of linear function approximation, and "neural PBO" when using neural networks;
- We moved several empirical results from the appendix to the main paper.

---

### Author Response · Authors · 2022-11-18
**Feedback on revision**

We are particularly grateful to the reviewers for their feedback that helped us improve the presentation of our work. We put significant effort to improve the paper and we would really like to know their opinion about the revised version. Before the deadline of tomorrow for sending revisions, we would appreciate knowing whether the reviewers are fine with the revision or if they would like some changes or additional content.

---

### Decision · Program_Chairs · 2023-01-20

**Decision:**

Reject

**Justification For Why Not Higher Score:**

Weaknesses in experimental methodology raise uncertainty in the effect of the proposed method

**Justification For Why Not Lower Score:**

N/A

**Metareview: Summary, Strengths And Weaknesses:**

This paper proposes an approximate Bellman operator challenging many fundamental assumptions in reinforcement learning. Perhaps inevitably when first presenting such fundamental ideas, many clarifying questions were raised by reviewers that the authors have worked hard to resolve in updates to the paper. However, after considering these revisions, the reviewers' questions on the experimental settings remained a major issue. A more complex environment (e.g. MuJoCo) is not strictly necessary in this instance, but the use of simpler environments should enable more extensive and clearer empirical results. Stronger empirical results could have both increased reviewer confidence in the significance of the proposed method and clarified the core contribution.